# High-Expansion Open-Cell Polylactide Foams Prepared by Microcellular Foaming Based on Stereocomplexation Mechanism with Outstanding Oil–Water Separation

**DOI:** 10.3390/polym15091984

**Published:** 2023-04-22

**Authors:** Dongsheng Li, Shuai Zhang, Zezhong Zhao, Zhenyun Miao, Guangcheng Zhang, Xuetao Shi

**Affiliations:** Key Laboratory of Macromolecular Science & Technology of Shaanxi Province, School of Chemistry and Chemical Engineering, Northwestern Polytechnical University, Xi’an 710072, China

**Keywords:** biodegradable polylactide, supercritical microcellular foaming, open-cell structure, stereocomplexation, oil–water separation

## Abstract

Biodegradable polylactic acid (PLA) foams with open-cell structures are good candidates for oil–water separation. However, the foaming of PLA with high-expansion and uniform cell morphology by the traditional supercritical carbon dioxide microcellular foaming method remains a big challenge due to its low melting strength. Herein, a green facile strategy for the fabrication of open-cell fully biodegradable PLA-based foams is proposed by introducing the unique stereocomplexation mechanism between PLLA and synthesized star-shaped PDLA for the first time. A series of star-shaped PDLA with eight arms (8-s-PDLA) was synthesized with different molecular weights and added into the PLLA as modifiers. PLLA/8-s-PDLA foams with open-cells structure and high expansion ratios were fabricated by microcellular foaming with green supercritical carbon dioxide. In detail, the influences of induced 8-s-PDLA on the crystallization behavior, rheological properties, cell morphology and consequential oil–water separation performance of PLA-based foam were investigated systemically. The addition of 8-s-PDLA induced the formation of SC-PLA, enhancing crystallization by acting as nucleation sites and improving the melting strength through acting as physical cross-linking points. The further microcellular foaming of PLLA/8-s-PDLA resulted in open-cell foams of high porosity and high expansion ratios. With an optimized foaming condition, the PLLA/8-s-PDLA-13K foam exhibited an average cell size of about 61.7 μm and expansion ratio of 24. Furthermore, due to the high porosity of the interconnected open cells, the high-absorption performance of the carbon tetrachloride was up to 37 g/g. This work provides a facile green fabrication strategy for the development of environmentally friendly PLA foams with stable open-cell structures and high expansion ratios for oil–water separation.

## 1. Introduction

Polymer foams with open-cell structures are a kind of porous material, with the gas phase as continuous microchannels and polymer phase as continuous bubble walls. Open-cell polymer foams have been applied in the field as catalyst supports, filters, tissue-engineering scaffolds, and porous electrodes, and for acoustic absorption due to their high surface area, high permeability and porosity [1,2,3,4]. For the fabrication methods of open-cell polymer foams, the gas foaming process could be considered the simplest one, using green gases instead of a solvent, compared with the tradition solvent-induced phase separation, salt leaching, and emulsion freeze-drying methods. With the further consideration of environmentally friendly polymer, biodegradable polylactide (PLA) foams with open-cell structures receive positive attention in the research and industrial fields of tissue engineering [5,6,7] and oil–water separation [8,9,10]. However, due to the intrinsic low melting strength and complex crystallization behavior of PLA especially under a supercritical condition, it is still a big challenge for the fabrication of PLA foams with uniform cell distribution and high expansion ratios [11]. In addition, the fabrication of PLA foam with an open-cell structure and high porosity is difficult because the cell growth with the connected channel can easily result in cell collapse.

In the literature, some of the reported strategies for the fabrication of open-cell polymer foams can be summarized into polymer-blending strategies with a second soft phase [12], using mixing blowing agents [13], creating a co-continuous microstructure [14]. The polymer blending of PLA with another biodegradable polymer such as Poly(butyleneadipate-co-terephthalate) (PBAT) [15,16,17] or Poly(butylene succinate) (PBS) [18,19] has already been reported, with an open-cell structure after foam blowing. However, the introduction of a second polymer would lead to the enhanced chain flexibility and decreased melting strength of PLA blends, while leading to poor compatibility and “sea-island” phase separation, which would lead to final open-cell foams with poor controllability and homogeneity and thus a narrow application area.

Compared with the limited choice of degradable materials, the addition of non-biodegradable polymers such as Polymethyl methacrylate (PMMA) [20,21], Polystyrene (PS) [22] and thermoplastic polyurethane (TPU) [23,24,25] can increase the melt strength of PLA and prepare PLA foams with higher expansion ratios. However, the non-biodegradable components would compromise the biodegradability and biocompatibility of PLA. In addition, the problem of poor compatibility of the blended system brings increased processing difficulties. Furthermore, there are some other strategies for improving PLA foamability, such as a filling modification with inorganic particles, and a copolymerization modification with Polycaprolactone (PCL) [26], Polyethylene glycol (PEG) [27], and Polyglycolic acid (PGA) [28]. For example, the added nanoparticles could act as cell-nucleating agents in the PLA composites, with consequentially enhanced foamability, while the nanoparticles are easily aggregated, make the foaming process difficult and cause non-uniform cell distribution. The copolymerization of PLA with other blocks would induce more complicated crystallization behavior and a consequentially narrow foaming process window.

In recent years, the unique PLA stereocomplexation mechanism has been promised to be one modification method that can be used due to the strong intermolecular forces between PLLA and PDLA, which lead to the higher melting temperature of the formed SC crystallites and their enhanced melting strength [29]. Therefore, improving the melting strength and regulating the crystallization property of PLLA based on the unique stereocomplex effect have been proposed as a new approach in the PLA foaming process. In our former work [30], the introduced SC-PLA crystallites could act as compatibilizers to improve the compatibility of PLLA with PBAT and also as cell nucleating agents for the foaming process. However, the current conventional SC-PLA often forms a closed-cell structure during physical foaming processes and has a low-expansion multiplier, which is because its crystallinity is too high and therefore limits the penetration of CO_2_, while the high melting strength is not conducive to material expansion. To sum up, a high melting strength is not favorable for the formation of an open-pore structure, and a low melting strength will cause cell aggregation and collapse during the foaming process, which has become a key scientific problem in the preparation of biodegradable open-pore PLA foam. In other words, the formation of stable and uniform micron-scale open-cell structures with suitable melting strength has become a major challenge to the fabrication of open-cell PLA materials and their applications.

In the literature, star-shaped polymers are multi-armed polymers with different numbers of branches being radiated from a central core, which present unique rheological, mechanical, and biomedical properties compared with those of linear polymer. Star-shaped PLA has been applied in the fields of chemistry and biomaterial engineering, based on the fact that it has more chain ends than linear polymers with the same molar mass do. Moreover, the special spatial structure of the star-shaped polymer has a low central density and gradually increasing density along the branching chain. In other words, the use of a special mesh structure of star-shaped molecules with a large free volume could be one strategy for the absorption of supercritical gases during the physical foaming process. In our former work, one hyperbranched epoxy resin [31] was introduced to the thermosetting epoxy and the system presented enhanced foamability due to the special hyperbranched epoxy structure and consequential gas diffusion ability. The above-mentioned star-shaped topological design and PLA stereocomplexation method could be combined for our target.

Herein, stellate PDLA (8-s-PDLA) with an eight-arm structure was designed and synthesized, and its spatial structure was utilized to enhance the penetration of physical foaming agents. SC-PLA was used to regulate the crystalline properties and melting strength of PLA. The low-molecular-weight PDLA did not have compatibility problems with the PLLA matrix, and could be blended in various ratios, thus maximizing the homogeneous and stable microscopic morphology of the pores. Through regulating the molecular weight and content of 8-s-PDLA, the microscopic morphology of PLA foam could be effectively controlled. The fabrication of environmentally friendly PLA-based foams with high expansion ratios and porosity, as well as outstanding adsorption properties, widens their application as load catalysis reactors, tissue engineering scaffolds, and oil–water separation materials.

## 2. Materials and Experiment

### 2.1. Experimental Materials

D-lactide was purchased from Shandong Daigang company, China. Tri-pentaerythritol (analytical purity, Macklin’s reagent) with eight hydroxy groups was used as an initiator for the ring-open polymerization of D-Lactide. Stannous octoate (analytical purity, Macklin’s reagent) was used as a catalyst. Commercial L-polylactide (PLLA-L170) was supplied by Purac. Co., Ltd. CO_2_ with a purity of more than 99.9%, obtained from Changte Air product Co., Ltd., was used as the blowing agent for microcellular foaming.

### 2.2. Synthesis of Star-Shaped PDLA with Eight Arms (8-s-PDLA)

The synthesis of 8-s-PDLA was carried out by the bulk ring-open polymerization of D-lactide with tri-pentaerythritol as an initiator and Sn(Oct)_2_ as a catalyst. In this work, the different feeding ratios of D-lactide to tri-pentaerythritol were set to 100:1, 50:1, 40:1 and 25:1, aiming to have a series of 8-s-PDLA of different molecular weights. Then, the Sn(Oct)_2_, with a weight percentage of about 0.1 wt%, of the above reaction system was added into the flask with a pipette, and this was followed by vacuuming and nitrogen inletting repeated in triplicate. The reaction flask was finally immersed in a 130 °C oil bath. After 48 h of polymerization, the products were dissolved in dichloromethane, dropped into ice-methanol for the precipitation and isolated by vacuum filtration, resulting in a white solid powder. The final product was obtained by repeating the above purification process three times, and further vacuum drying.

### 2.3. Preparation of PLLA/8-s-PDLA

PLLA/8-s-PDLA sheets were prepared by the solution casting method and mini-injection molding. PLA pellets were dried at 60 °C for 12 h in oven. The PLLA/8-s-PDLA blend was firstly prepared in the dichloromethane, with different 8-s-PDLA contents of 3 wt%, 5 wt% or 10 wt%. A series of PLLA/8-s-PDLA specimens with stable dimensions were fabricated by adding the casted films mentioned above into a mini-injection molding machine with a processing temperature of 200 °C.

### 2.4. Supercritical Carbon Dioxide Microcellular Foaming

The supercritical foaming of PLLA-based blends was carried out in an autoclave (Anhui Kemi Technology Co., Ltd. Anhui Province, China). The batch foaming process involved a gas saturation under a certain temperature and pressure and the consequentially rapid pressure relief for foaming, according to the cell nucleation and cell growth, respectively. In this work, the saturation temperature was set from 115 °C to 135 °C and the saturation pressure was from 16 Mpa to 22 Mpa, with the holding time being from 2–4 h.

### 2.5. Characterization and Analysis

The chemical structure of synthesized 8-s-PDLA was characterized by a nuclear magnetic resonance spectrometer (NMR) (Bruker Advance 400 MHz, Karlsruhe, Germany). Deuterated chloroform was chosen as solvent for the 8-s-PDLA. The number-average molecular weight of the synthesized 8-s-PDLA could be derived from the calculation of the ratio of the characteristic peak area of the end groups to the ones of the repeating unit. A series of synthesized 8-s-PDLA was characterized in order to discuss the influence of the feeding ratio on the final molecular weight.

The characteristic temperature of 8-s-PDLA was determined by DSC (DSC 3 from Mettler-Toledo, Zurich, Switzerland). About 5 mg of 8-s-PDLA in a 40 μL aluminum crucible was tested by the following thermal program. The program involved firstly heating the material from 20 °C to 200 °C, maintaining it at 200 °C for five minutes to eliminate the thermal history of the material, then cooling it down to 20 °C (at the isotherm for five minutes), and finally ramping it up to 200 °C again. The thermal rate was fixed to 10 °C/min, and nitrogen was used as the protective gas.

A gel permeation chromatography (GPC, Waters 1515, Waltham, MA, USA) measurement was also carried out for the determination of the molecular weight and molecular weight distribution of the 8-s-PDLA, with tetrahydrofuran as the mobile phase.

The thermal properties and crystallization behavior of the PLLA/8-s-PDLA blends were characterized by DSC. The DSC thermal program of PLLA/8-s-PDLA was set by the ramp-up procedure from 20 °C to 230 °C, and then was set to be isothermal for 5 min as the first heating procedure, followed by cooling it back to 20 °C and applying a second heating procedure to reach 230 °C. The heating rate was 10 °C/min and nitrogen was the protective gas.

The crystallization behavior of PLLA/8-s-PDLA was also characterized by XRD (XRD-7000, Shimadzu, Kyoto, Japan). XRD test specimens were prepared by the solution volatilization method, with the same concentration of PLA in the dichloromethane solution to ensure the same crystallization growth history.

The crystalline morphology of the PLLA/s-PDLA system was observed by a hot-stage polarizing microscope test. Different PLLA/s-PDLA solutions of the same concentration were prepared, the same volume of drops was measured on a coverslip with a pipette, and the test samples were obtained after the complete evaporation of dichloromethane. The prepared samples were placed on a polarizing microscope equipped with a heating table, the specimens were heated above 230 °C for 5 min to ensure the complete melting of the specimens (the elimination of the thermal history) and rapidly cooled down to 130 °C to observe the nucleation and growth process of isothermal crystallization, and the process and results of crystallization were recorded with a camera.

The rheological properties of PLLA/8-s-PDLA were determined by a rotational rheometer (RH20, TA, New Castle, DE, USA). The related complex viscosity, loss modulus, energy storage modulus and loss factor of PLLA/8-s-PDLA versus the angular frequencies from 100 rad/s to 1 rad/s were determined under the temperature conditions of 190 °C and 200 °C, respectively. Furthermore, the angular frequency of 10 rad/s was fixed to investigate the rheological properties of the specimens versus the temperature ranging from 180 °C to 230 °C.

The microscopic morphology of PLA foam was characterized by SEM (VEGA 3 LMH, Brno, Czech Republic). The cross-section of the fractured surface was sprayed with gold and observed by SEM. The average cell size and cell density of PLLA-based foam was analyzed by the photo statistics of the corresponding SEM observation by the Image Pro Plus software with at least 100 bubble size data points, in accordance with the following equations, Equations (1) and (2). The bubble size calculation formula is as follows.
(1)$$\phi =\frac{\sum d_{i}n_{i}}{\sum n_{i}}$$
(2)$$N_{f}={\left(\frac{nM^{2}}{A}\right)}^{\frac{3}{2}}\cdot \left(\frac{\rho_{s}}{\rho_{f}}\right)$$
where *φ* is the average cell size, $d_{i}$ is the cell diameter, and $n_{i}$ indicates the number of cells. *N*_f_ is the cell density, *n* is the cell number in the SEM image, *M* denotes the magnification, *A* denotes the area of the statistical image, *ρ*_s_ is the density of the solid, and *ρ*_f_ is the density of the foam.

The foam density of PLLA/8-s-PDLA foams is that of all open-cell foams, but with a dense molten layer on the surface. The foam density was measured according to ASTM D792-13 following Equation (3):(3)$$\rho_{f}=\frac{m_{1}}{m_{1}-m_{2}}\cdot \rho_{w}$$
where *ρ*_f_ is the density of the foam, *m*_1_ is the peak dry weight of the foam, *m*_2_ is the mass of the foam in the water, and *ρ*_w_ is the density of pure water, for 0.9975 g/cm^3^.

The expansion ratio of the PLLA/8-s-PDLA foams was determined by the ratio of the density from before foaming to after foaming.

## 3. Results and Discussion

### 3.1. Characterization of Synthesized 8-s-PDLA

The chemical structure of the synthesized 8-s-PDLA was characterized by NMR, as shown in Figure 1. The structure and number-average molecular weight of stellate PLA were determined by comparing the characteristic hydrogen protons on tri-pentaerythritol with those on D-LA.

The structure of tripentaerythritol can be determined from the integrated area of the peak shapes of peaks 5 and 6 in Figure 1. The hydroxyl group located at the end of the PDLA chain causes the chemical shift of the hydrogen atoms on the lactic acid structural unit at the end of the f-molecular chain to move to the lower field, i.e., peaks 1 and 3. The number-average molecular weight of a single-star arm can be calculated by comparing the area of peak 1 and peak 3, provided that only one terminal hydroxyl group is determined for each molecular chain. Since the end hydroxyl group is an active hydrogen, it is not discussed too much here; the characteristic hydrogen on the side of the methyl structure of the PDLA main chain is affected less by the end-hydroxyl group, the chemical shift of the methyl hydrogen at different positions is less, and the overlap phenomenon is not conducive to quantitative analysis, so it is not discussed too much here. By comparing the characteristic hydrogen on the nucleus’ structure and the characteristic hydrogen on the lactic acid’s structure at the end of the chain, the number of star arms of the star-shaped molecule can be analyzed. The specific analysis is as follows:

The resonance peak of the methine proton in the repeating unit of the star-shaped PDLA was observed from 5.0 to 5.5 ppm. The proton peak attributed to the terminal methine group of the PLA shifted from 4.25 to 4.5 ppm. There were the characteristic peaks of methyl proton in the repeating unit and the end group, marked as 2 and 4 from 1.25 to 2.0 ppm, respectively. The proton peaks (peak 5) from 4.0 to 4.25 ppm were the methylene proton in the side group of tri-pentaerythritol, and the proton peak 6 at 3.3 ppm was the methylene proton in the main chain. Based on the calculation, the integral area ratio of three proton peaks (peak 3: peak 5: peak 6) was 1:2:1, which confirmed the successful synthesis of star-shaped PDLA. The average-number molecular weight of the star-shaped PDLA could be roughly calculated by the integral ratio of the proton peak of the methyl at the repeat unit to that at the end group. Based on the different feeding ratios of monomer to initiator, the different ratios of peak 1 to peak 3 are obvious, as shown in the left of Figure 1. The calculated molecular weights are listed in Appendix A and are compared with those determined by GPC measurements. Then, four different 8-s-PDLA foams with different molecular weights were marked as 8-s-PDLA-8K, -13K, -15K and -39K.

The DSC curves of the second heating procedure of 8-s-PDLA with different molecular weights are shown in Figure 2. The melting temperature of 8-s-PDLA is highly dependent on its molecular weight and molecular weight distribution. The melting point of 8-s-PDLA increases with increasing molecular weight, and the melting range is wider with an increasing molecular weight distribution. For example, the melting temperature of 8-s-PDLA-39K was about 157.5 °C. The DSC results indicated the trends of the melting point versus molecular weight, which are consistent with the GPC results and molecular weight calculated by NMR, as shown in Appendix A.

With the purpose of enhancing the foamability of PLLA-based blends, the crystallization behavior and the related melting strength of PLLA/8-s-PDLA blends are key issues. Therefore, the influence of the synthesized 8-s-PDLA with different molecular weights and different 8-s-PDLA contents on the crystallization and rheological properties of PLLA/8-s-PDLA were firstly investigated.

For the crystallization behavior of PLLA/8-s-PDLA, some issues are critically important, such as the star-shaped topology of 8-s-PDLA and the stereocomplexation between PLLA and 8-s-PDLA. As shown in Figure 3, the stereocomplex crystallite stereocomplexation crystallite had a second melting peak (*T*_m,SC-PLA_) at about 190 °C in the PLLA/8-s-PDLA blends, in contrast to the pure PLLA at 170 °C. This indicates that the PLA stereocomplexation mechanism also works between PLLA and synthesized 8-s-PDLA. The corresponding thermal parameters of the glass transition temperature, melting temperature of homocrystallized crystallites (*T*_m,HC-PLA_) and *T_m,SC-PLA_* are recorded in Appendix A. In addition, the melting temperature, *T_m,SC-PLA_*, increases with the increase in the molecular weight of 8-s-PDLA (Figure 3c), and its enthalpy increases with the increase in the content of 8-s-PDLA (Figure 3a).

The influences of the molecular weight or the content of 8-s-PDLA on the crystallization behavior of PLLA are further elaborated by the DSC curves and XRD patterns in Figure 3. The DSC curves of the second heating process for pure PLLA present a wide and flat cold crystallization peak in Figure 3a,c, which is due to its intrinsic slow crystallization rate. On the other hand, with the addition of 8-s-PDLA and the formation of SC-PLA, the PLLA/8-s-PDLA blend exhibits obvious sharper cold crystallization peaks, which indicates an enhanced crystallization rate. In other words, the addition of 8-s-PDLA to PLLA can effectively reduce the onset temperature of the cold crystallization of PLLA. Additionally, this effect is more obvious when the molecular weight of PDLA is lower; for example, the cold crystallization temperature of the PLLA/8-s-PDLA-13K can be lowered to below 115 °C. The effect of 8-s-PDLA on the crystallization behavior of PLLA is complex. As shown in Figure 4a, the cold crystallization peak is at about 106 °C for the PLLA/8-s-PDLA-13K-5%, which is lower than that of PLLA/8-s-PDLA-13K-3% or PLLA/8-s-PDLA-13K-10%. In other words, the homocrystallization of PLLA could be enhanced with a suitable amount of 8-s-PDLA, while a higher 8-s-PDLA content would lead to slower cold crystallization due to the confinement effect of the existing SC-PLA crystallites as a physical cross-linking point.

As shown in Figure 3b,d, the XRD patterns of the PLLA and PLLA/8-s-PDLA blends were compared, mainly considering the molecular weight and contents of s-PDLA. Similar results can be found in the XRD patterns shown in Figure 3b; the addition of low-molecular-weight 8-s-PDLA-13K to PLLA resulted in enhanced homocrystallization peaks (the characteristics peaks being at 16.7° and 19.1°), while the formation of a large amount of SC-PLA has corresponding peaks at 11.9°. With the increase in 8-s-PDLA contents from 3 wt% to 10 wt%, the crystalline peaks of both HC-PLA and SC-PLA firstly increase and then decrease, as shown in Figure 4d, which again indicates that a large amount of the existing SC-PLA acted as a physical cross-linking point and prevented molecular chain mobility instead of the nucleation effect.

The reasons for the effect of star-shaped PDLA on the crystallization behavior of PLLA systems can be explained by the following two aspects. On the one hand, the SC-PLA crystallites could act as nucleation sites to increase the nucleation rate of PLA crystallization; on the other hand, the physical cross-linking network of SC-PLA could constrain the mobility of the molecular chains of the PLA and thus reduce the crystallization rate of the PLLA. In order to further confirm the results of DSC and the XRD measurements, the direct isothermal crystallization of PLLA and PLLA/8-s-PDLA blends with different molecular weights and contents of star-shaped PDLA was investigated by POM measurements and is shown in Figure 4 and Appendix A. When isothermally treated at 130 °C for 3 min, PLLA/8-s-PDLA-13K-5% presented more spherulites compared to pure PLLA or PLLA with more 8-s-PDLA-13K-10%. This indicates that the star-shaped SC-PLA formed acted as a nucleating agent for homocrystallization. For the PLLA/8-s-PDLA-13K-10% blends, the spherulite density was much lower than that of the other blends, while the diameter of final spherulites was also smaller than that of the other blends after isothermal heating for 30 min, which confirms that the restriction to the PLLA’s molecular mobility was due to the formation of SC-PLA crystallites.

Synthesized 8-s-PDLA with low molecular weights and special spatial structures could exhibit a plasticizing effect on a PLLA matrix and could be beneficial to the growth of PLA crystallites; star-shaped PDLA could form SC-PLA with the PLLA matrix, resulting in the promoted nucleation of the PLLA crystallite [32] while slowing crystallite growth [33]. The final crystallization properties in PLLA/8-s-PDLA systems are determined by these three factors simultaneously. When the molecular weight and content of 8-s-PDLA are high, the effect of SC-PLA on restraining molecular chain movement is greater than its effect on promoting molecular chain movement.

The above-mentioned influence of 8-s-PDLA on the crystallization behavior of PLLA-based blends further affects the foaming behavior. It is well-known that a high crystallinity and fast crystallization rate would lead to the slow diffusion and lower saturation of supercritical carbon dioxide, while low crystallinity is not beneficial to enhancing PLLA’s melting strength, which makes it difficult to form a final form with a high expansion ration. Melting strength plays a crucial role in the physical foaming of PLA, especially since the preparation of open-cell foams with high-expansion multiplicity requires a suitable melting strength.

The rheological properties of PLLA and PLLA/8-s-PDLA at 190 °C are compared in Figure 5a1–a3. The addition of 8-s-PDLA has a significant effect on the rheological properties of PLLA 190 °C. The storage modulus G’ of PLLA/8-s-PDLA with the addition of 8-s-PDLA exhibits a nearly linear trend with increasing frequency (Figure 5a1). Meanwhile, the addition of 8-s-PDLA results in greatly enhanced complex viscosity (Figure 5a3). This indicates that the addition of 8-s-PDLA effectively improves the melting strength of PLA blends. The reason for this phenomenon is assigned to the physical crosslinking point of the formed SC-PLA reserved at 190 °C, which presents the dependence on the molecular weight of 8-s-PDLA and cross-linking density of SC-PLA between PLLA and 8-s-PDLA. At low frequencies, the physical cross-linked network would lead to a more obvious increase in viscosity, while at high frequencies, the strong shearing effect will simultaneously break the entangled molecular chain and also the physical cross-linked network, leading to a rapid drop in viscosity and a viscosity of a similar value to that of pure PLLA. 

In contrast, when the PLLA/8-s-PDLA blends were measured at 200 °C, the rheological results shown in Figure 5b were strongly related to the molecular weight of the 8-s-PDLA. The storage modulus and complex viscosity of PLLA/8-s-PDLA-13K and PLLA/8-s-PDLA-15K (a lower molecular weight) are at the same level with those of pure PLLA, while those of PLLA/8-s-PDLA-39K (high molecular weight) are much higher than those of pure PLLA. The reason can be found in Figure 3c, which is that the melting point of PLLA/8-s-PDLA-39K is about 210 °C. This is consistent with the results in Figure 5a, which confirm the dominating effect of SC-PLA as a physical cross-linking point for the enhancement of the melting strength. In addition, the formation of SC-PLA caused by the strong intermolecular forces is strongly related to the molecular weight of 8-s-PDLA, while the breaking of well-ordered SC-PLA by the destruction of the intermolecular forces is dominated by temperature.

Figure 5c1–c3 shows the change in the rheological properties of the PLLA/8-s-PDLA blends versus the temperature, and that the storage modulus G’ and complex viscosity of the PLLA/8-s-PDLA blends were higher than those of the pure PLLA at 180 °C. When the temperature further increased, the related viscosity of PLLA/8-s-PDLA-13K started to fluctuate greatly at about 191 °C. Similarly, the rheological curves of PLLA/8-s-PDLA-15K present a fluctuation at a temperature of 195 °C, while showing that the PLLA/8-s-PDLA-13K blend exhibited a repaid decrease in viscosity at around 210 °C. This phenomenon was caused by the melting of SC-PLA induced by the formation of 8-s-PDLA, and those transition temperatures perfectly match the results in Figure 3, which depend on the molecular weight. The strength of SC-PLA formed with low-molecular-weight 8-s-PDLA is low and its distribution is narrow, making the viscosity of the system change drastically in a short time and forming unstable changes in the rheological curve. In contrast, the SC-PLA induced by PDLA of a larger molecular weight was stronger, and a relative long temperature range would be necessary for the complete melting of the formed SC-PLA.

The addition of star-shaped PDLA obviously improves the melting strength of PLA, which could further affect the consequential microcellular foaming behavior [29]. More importantly, the molecular weight of star-shaped PDLA plays an important role in regulating the viscosity of a system and, further, microcellular morphology.

### 3.2. Foaming Behavior of PLLA/8-s- PDLA Blends during Microcellular Foaming

The influence of added 8-s-PDLA and of the consequentially formed SC-PLA on the crystallization and rheological properties of the PLLA-based blend is complex, which greatly affects its further foamability during supercritical CO_2_ foaming. The PLLA/star-shaped PDLA blends were a homogeneous system, but the reserved SC-PLA crystallites could act as cell nucleation sites during the foaming process, which constrained the gas absorption but facilitated cell nucleation. In addition, the influence of the blending composition and foaming condition (temperature and pressure) on the foaming behavior of the material are critical for determining the microcell morphology and microproperties of foams.

Figure 6 shows the SEM images of the pure PLLA and PLLA/8-s-PDLA blends with 8-s-PDLA of different molecular weights. Differently from the traditional close-cell structure of the pure PLLA sample in Figure 6a, all those PLLA/8-s-PDLA foams exhibited an open-cell structure with high expansion ratios and much more uniform cell distribution. In detail, the PLLA/8-s-PDLA foam with the lower-molecular-weight 8-s-PDLA-13K presented a higher open-cell ratio and expansion ratio of 14.5, with an average cell size of 23.6 um, as shown in Figure 6b. For the PLLA/8-s-PDLA-39K foam with a higher molecular weight, the related average cell size was 14.6 um, as shown in Figure 6d, and the expansion ratio was about 7.8.

The foaming mechanism of PLLA/8-s-PDLA blends is strongly related to the molecular weight of the star-shaped PDLA and the formation of SC-PLA, as shown in Figure 7. Considering the different steps of supercritical microcellular foaming, the introduced star-shaped SC-PLA crystallites could act as gas reservoirs and cell nucleation sites. Differently from linear commercial PDLA, synthetic 8-s-PDLA has a relatively low molecular weight and much more free volume, as well as a higher physical cross-linking density caused by the eight arms participating in PLA sterecomplexation. Therefore, the totally compatible 8-s-PDLA and the formed SC-PLA could induce more cell nucleation, which followed the cell growth towards the weaker phase. In this PLLA/8-s-PDLA blend system, the weaker phase of 8-s-PDLA could not afford a high pressure during foam growth, and was followed by the formation of open-cell structure. Meanwhile, the existing SC-PLA with strong intermolecular forces could avoid further cell coalescence and collapse to solidify the final foam dimension. Therefore, the above-mentioned PLLA/8-s-PDLA presented an open-cell structure and cell distribution with good uniformity; more importantly, the stability of the cell structure resulted in a high expansion ratio.

Meanwhile, the influence of the amount of 8-s-PDLA-13K added to the PLLA on the final cell morphology was investigated, as shown in Figure 8. With the increasing contents of 8-s-PDLA-13K, the SEM images of foam morphology exhibit the increase in the average cell size and expansion ratio, while exhibiting the decreased foam density, which was about of 0.09 g/cm^3^. A greater amount of 8-s-PDLA meant that the more dispersed SC-PLA particles acting as physical cross-linking points not only enhanced the melting strength but also benefited gas absorption and cell nucleation. During ell growth, the gas broke up the 8-s-PDLA-13K, leading to a bigger open-cell size, while the expansion ratio could be increased and be well-kept due to the higher density of the SC-PLA. In other words, 8-s-PDLA of a smaller molecular weight would be more critical in the formation of an open-cell structure, while -PDLA of a higher molecular weight would be more effective at retaining the the expansion stability of the final foam. As shown in Figure 8d, the PLLA blends with 1% 8-s-PDLA-13K and 2% 8-s-PDLA-39 presented a cell size of about 12.4 μm and expansion ratio of about 7.0, but presented the highest cell density, which was about 5.26 × 10^9^. The main advantage of the enhanced foamability of PLLA-based blends is the small requirement of only 1 wt% synthetic 8-s-PDLA without any incompatibility or costs to its outstanding biodegradability.

In order to fully demonstrate the effect of temperature and pressure on the pore shape of the PLLA/8-s-PDLA foam during the holding process, and to facilitate the full penetration of carbon dioxide into the PLLA/8-s-PDLA material at lower temperatures and pressures, a thinner specimen was chosen for supercritical holding. The foaming processing conditions of PLLA/8-s-PDLA-15K were also modified further to optimize the conditions, as shown in in Figure 9 and Figure 10. PLLA/8-s-PDLA-15K-5% foams prepared under different foaming conditions exhibited a stable open-cell structure without cell coalescence and collapse in the wider temperature range from 115 °C to 135 °C, as shown in Figure 9, and exhibited a pressure range from 15MPa to 22Mpa, as shown in Figure 10. When the pressure was kept at 20 MPa, the average cell size of PLLA/8-s-PDLA-15K-5% increased from 18.0 to 26.5, and the related expansion ratio increased from 6.1 to 9.0, when the temperature increased from 110 to 135 °C. An even higher temperature would have caused cell collapse, while a lower temperature would have made it difficult to form a complete open-cell structure. With the increase in the foaming pressure, the average cell size of PLLA/8-s-PDLA-15K-5% decreased from 25–26 μm to 20–21 μm, at a fixed temperature of 120 °C and a pressure from 16 MPa to 22 MPa. An increase in pressure would cause higher-capacity adsorption for the adsorption of gas and would cause more gas nuclei to be formed in the saturation process. Therefore, the cell density increases and the average cell size decreases. The details of average cell size, cell density, expansion ratio and cell density are listed in Appendix A.

By controlling the foaming condition and the added 8-s-PDLA in the PLLA matrix, the microstructure and macro-density of PLLA/8-s-PDLA could be regulated, and the detailed information of the related foams are listed in Table 1. For example, a reticulated-pore PLLA/8-s-PDLA-13K-5% foam with an expansion ratio of about 20 and a pore size of about 20 μm could be formed at 130 °C and 20MPa; a PLLA/8-s-PDLA-13K-5% foam of a cell size of 61 μm and an expansion ratio of 25 or more could be formed under the foaming conditions of 130 °C and 17 MPa.

PLLA/8-s- PDLA foams prepared by microcellular foaming with a high-porosity open-cell structure and high expansion ratio are reported in the literature as the basic requirements for the fabrication of materials to be applied in oil–water separation. Herein, the adsorption capacity of the prepared PLLA/8-s- PDLA foams was measured using CCl_4_ and silicone oil as two examples. The direct digital pictures of the silicone oil absorption processes undergone by the PLLA/8-s-PDLA blends are shown in Figure 11, and the absorption process could be finished in 10 s. In detail, a series of PLLA/8-s- PDLA foams, as discussed above, were compared for their absorption capacity for the adsorption of either silicone oil or CCl_4_, as shown in Figure 11a,b, respectively. All those samples presented saturated absorption within 10 s, which is very fast and convenient. For the capacity of adsorbing silicone oil, the PLLA/8-s-PDLA-13K foam prepared under a pressure of 17 MPa presented the highest absorption value of 18.3 g/g. Similarly, this system also had the highest absorption capacity for CCl_4_ of about 37.0 g/g. These values are also comparable to those found in the literature for the absorption capacity of foams with open-cell structures. This work presents the result of higher absorption, and the working mechanism is also considered. The highly networked cell structure of PLLA/8-s-PDLA foam provides it with an excellent three-dimensional spatial structure and large specific surface area, which enable oil substances to fully contact the foam and penetrate into the interior quickly and the saturation adsorption capacity to be reached in a short time. The measured absorption values are compared with the related SEM images of foam morphology. It is obvious that the higher the porosity of PLLA-based foams, the higher the absorption capacity for silicone oil or CCl_4_.

From Figure 11 and Figure 12, it can be seen that foam with PLLA/8-s-PDLA has an extremely strong adsorption capacity for CCl_4_ and silicone oil, reaching more than 90% of the maximum adsorption capacity in about 10 s. The larger the pore size of the foam, the higher the expansion multiplier, and the stronger the adsorption capacity. The highly networked pore structure provides PLLA/8-s-PDLA foam with an excellent three-dimensional spatial structure and large specific surface area, which enable oil substances to fully contact the foam and penetrate into the interior quickly and the saturation adsorption capacity to be reached in a short time. From Figure 12, it can be seen that PLLA/8-s-PDLA-13K foam exhibited the strongest adsorption capacity, and that the amount of adsorption of silicone oil could reach 18.26 g/g and 37 g/g for CCl_4_. The adsorption capacity of the foam was compared with the results in the literature on PLA open-cell foams by blending with it other polymers, as shown in Table 2, which presents the outstanding adsorption capacity. Traditional modified PLLA with a large amount of PBS and PBAT is difficult to prepare by the physical foaming method because of the incompatibility and requirement of adding a large amount of a second polymer. In contrast, this strategy for the fabrication of PLLA/8-s-PDLA microcellular foam is facile with high efficiency and stability.

## 4. Conclusions

In this work, a series of eight-armed star-shaped PDLA foams of different molecular weights were synthesized by ring-opening polymerization using tri-pentaerythritol as an initiator, which were further added to commercial PLLA as modifiers. The crystallization behavior and rheological properties of the PLLA/8-s-PDLA blends were investigated, considering the molecular weight of 8-s-PDLA and its contents in the PLLA matrix. The results indicated that the homocrystallization behavior of PLLA was greatly enhanced by the reserved SC-PLA crystallites acting as nucleation sites because of the unique stereocomplexation mechanism between PLLA and 8-s-PDLA. In addition, PLA stereocomplexation with only small amount of 8-s-PDLA was confirmed to enhance rheological properties, due to effect of the physical cross-linking point, without inducing any phase separation when the PLA was blended with other polymers.

The microcellular foaming behavior of PLLA/8-s-PDLA was investigated by considering the following aspects: the molecular weight of 8-s-PDLA, the added contents of 8-s-PDLA and the foaming conditions. Differently from the traditional closed-cell structure, all the PLLA/8-s-PDLA foams exhibited an open-cell structure with a high interconnection ratio, high expansion ratio and uniform cell distribution. The influence of the addition of 8-s-PDLA on the formed SC-PLA plays critical roles during microcellular foaming. The totally compatible SC-PLA particles, taken as physical cross-linking points, acted as gas reservoirs and cell nucleation sites, facilitating cell nucleation and cell growth. During the cell growth process, the gas preferred to escape towards the weaker phase, leading to the formation of an open-cell structure. Meanwhile, the enhancement of rheological properties was critical for the solidification of the interconnected structure without cell collapse, resulting in a high expansion ratio of up to 24 and a low foam density of around 0.05 g/cm^3^.

This work also investigated the potential application of the prepared open-cell PLA foams as biodegradable alternatives to the tradition foams used in the field of oil–water separation. The results reported that the high-adsorption capacity for CCl_4_ of PLLA/8-s-PDLA-13K foams could be about 37.0 g/g. This work provides a new strategy for the facile fabrication of environmentally friendly PLA foams without compromising its outstanding biodegradability based on the small amount of star-shaped PDLA. The resulting high porosity and high expansion ratio greatly widens its application not only in the field of oil–water separation but also as a tissue engineering scaffold, and so on.

## Figures and Tables

**Figure 1 polymers-15-01984-f001:**
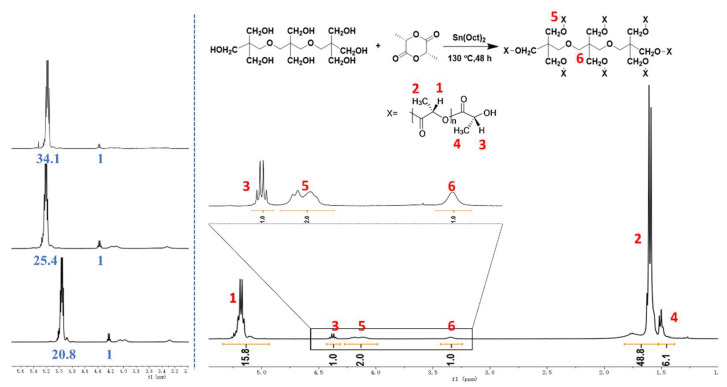
^1^H NMR spectra of synthesized star-shaped PDLA.

**Figure 2 polymers-15-01984-f002:**
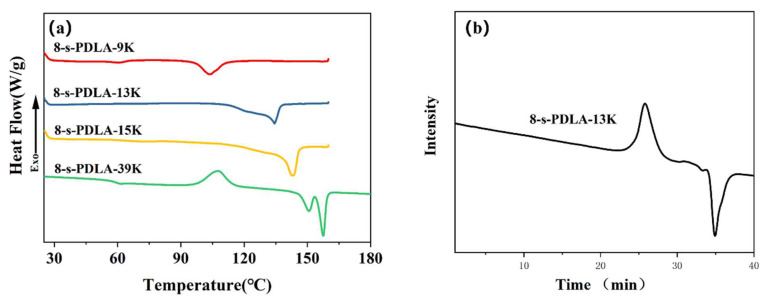
(**a**) DSC heating curves of 8-s-PDLA with different molecular weights. (**b**) GPC curve for 8-s-PDLA-13KCharacterization of PLLA/8-s-PDLA Blends.

**Figure 3 polymers-15-01984-f003:**
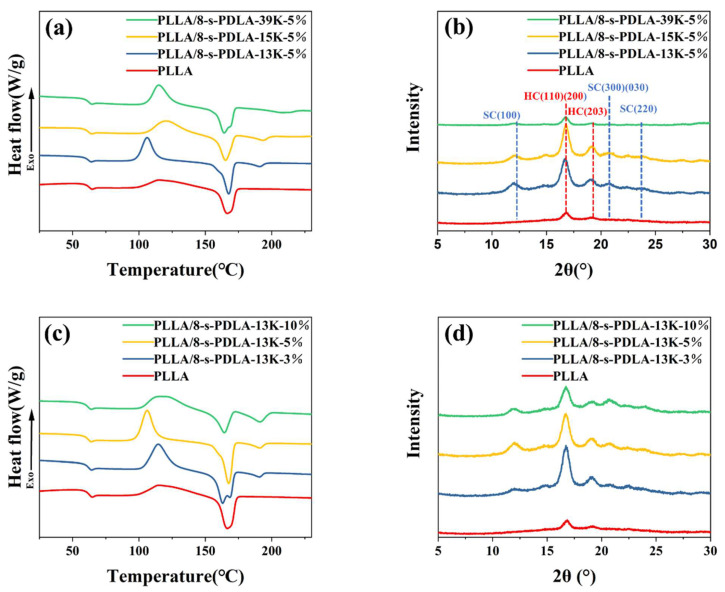
(**a**) DSC second heating curves and (**b**) XRD patterns of PLLA-based blends with different contents of 8-s-PDLA-13K; (**c**) DSC second heating curves and (**d**) XRD patterns of PLLA-based blends with 8-s-PDLA-13K of different molecular weights.

**Figure 4 polymers-15-01984-f004:**
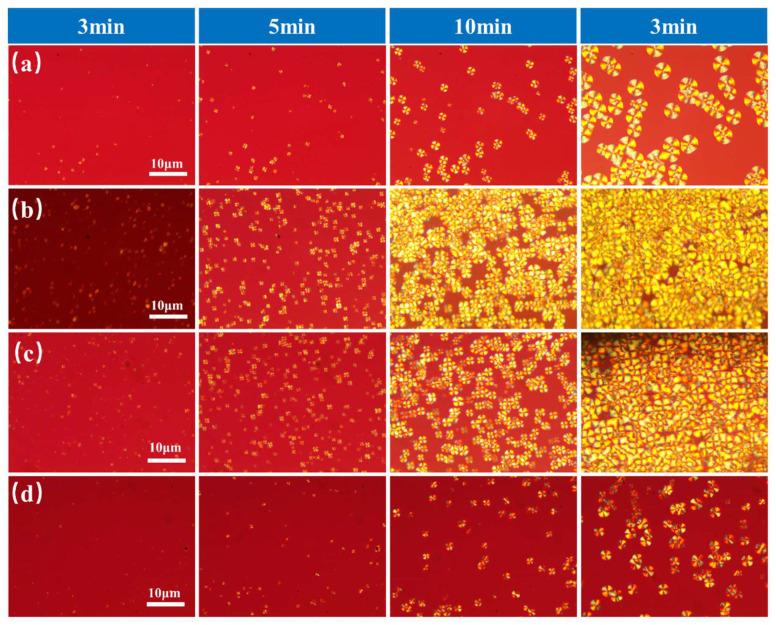
POM photographs of isothermal PLLA/8-s-PDLA-13K blends crystallized at 130 °C: (**a**) pure PLLA, (**b**) PLLA/8-s-PDLA-13K-3%, (**c**) PLLA/8-s-PDLA-13K-5%, and (**d**) PLLA/8-s-PDLA-13K-10%.

**Figure 5 polymers-15-01984-f005:**
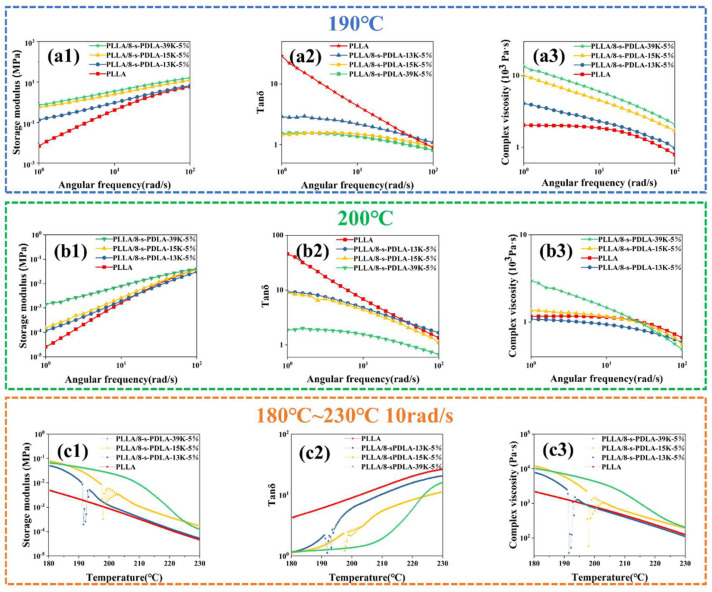
Frequency dependence of storage modulus (**a1**), loss tangent angle (**a2**) and complex viscosity (**a3**) of PLLA with 8-s-PDLA of different molecular weights blended at 190 °C; Frequency dependence of storage modulus (**b1**), loss tangent angle (**b2**) and complex viscosity (**b3**) of PLA with 8-s-PDLA of different molecular weights blended at 200 °C; temperature dependence of storage modulus (**c1**), loss tangent angle (**c2**) and complex viscosity (**c3**) of PLA with 8-s-PDLA of different molecular weights at fixed frequency of 10 rad/s from 180 to 230 °C.

**Figure 6 polymers-15-01984-f006:**
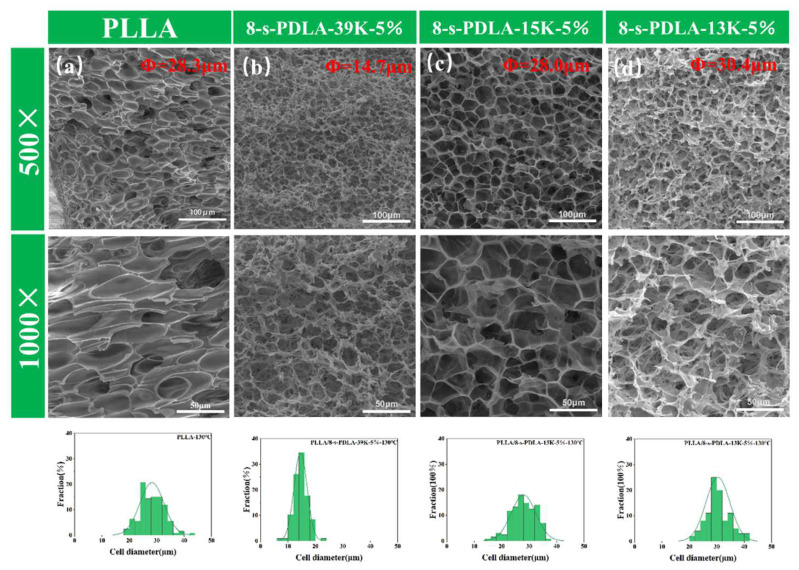
SEM images of cell morphology of: (**a**) pure PLLA; (**b**) PLLA/8-s-PDLA-13K-5%; (**c**) PLLA/8-s-PDLA-15K-5%; and (**d**) PLLA/8-s-PDLA-39K-5%. The foaming condition was kept the same at 130 °C with a pressure of 20 MPa for 4 h, with following fast pressure release.

**Figure 7 polymers-15-01984-f007:**
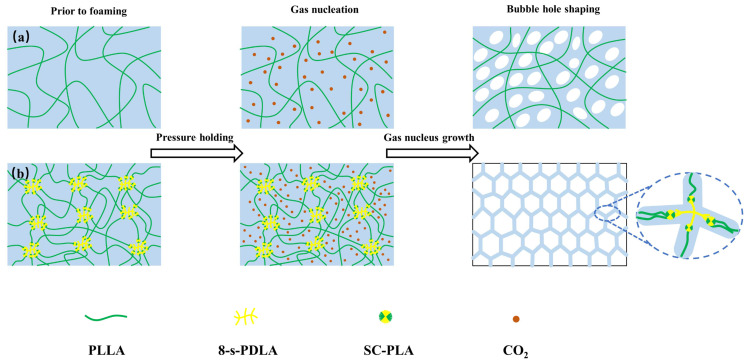
Schematic illustration of microcellular foaming of (**a**) pure PLLA and (**b**) PLLA/8-s-PDLA considering the effect of SC-PLA on cell nucleation and cell growth.

**Figure 8 polymers-15-01984-f008:**
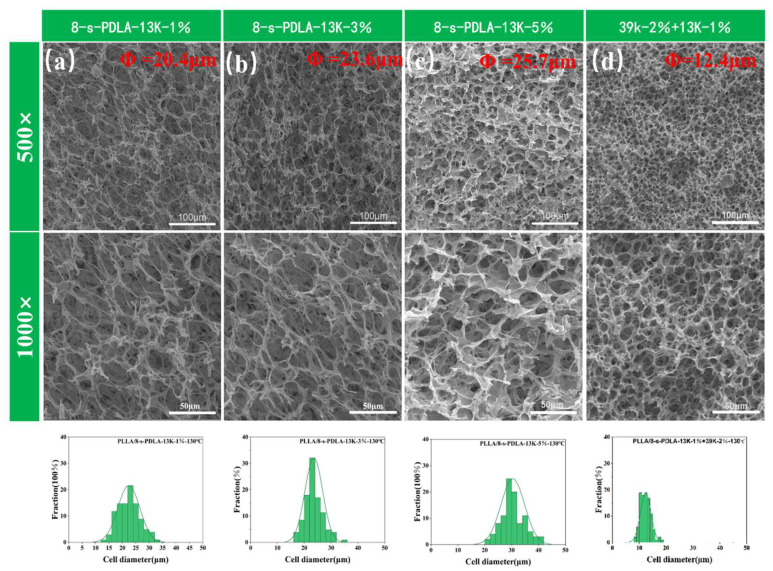
SEM images of cell morphology of: (**a**) PLLA/8-s-PDLA-13K-1%, (**b**) PLLA/8-s-PDLA-13K-3%, (**c**) PLLA/8-s-PDLA-13K-5% and (**d**) PLLA/8-s-PDLA-13K-1% + 39K-2% (the foaming condition was kept the same at 130 °C with a pressure of 20 MPa for 4 h, with following fast pressure release).

**Figure 9 polymers-15-01984-f009:**
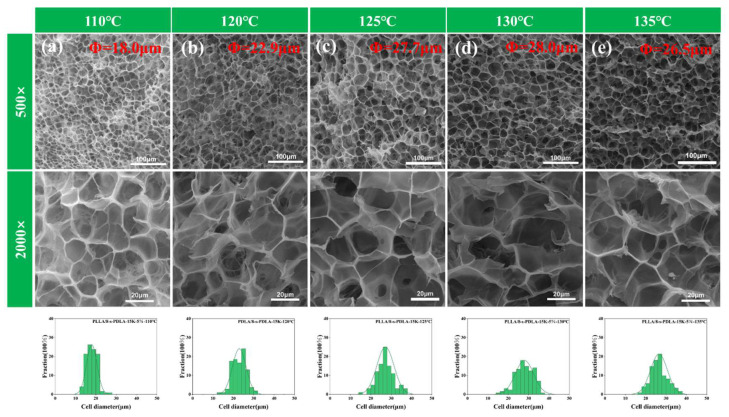
SEM images of cell morphology for PLLA/8-s-PDLA-15K-5% prepared by different foaming temperatures: (**a**) 110 °C, (**b**) 120 °C, (c)125 °C, (**d**) 130 °C and (**e**) 135 °C when the pressure was fixed at 20 MPa for 2 h.

**Figure 10 polymers-15-01984-f010:**
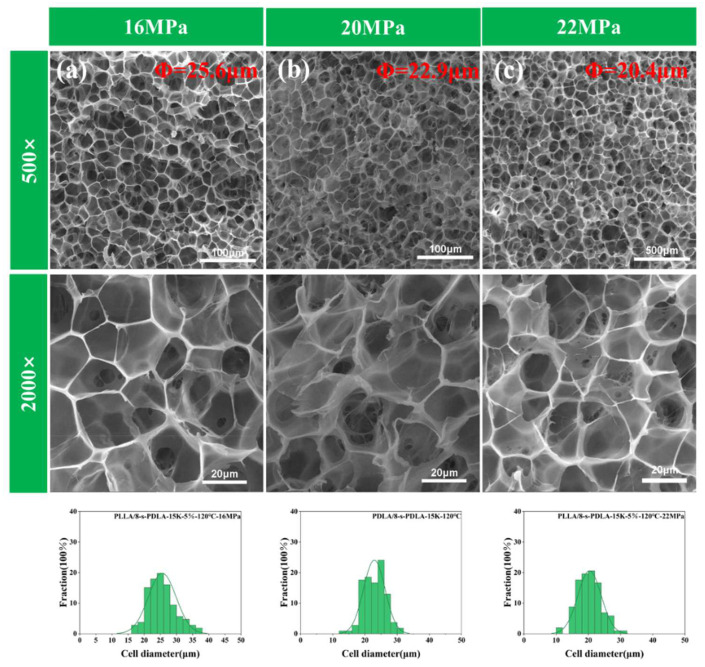
SEM images of cell morphology of PLLA/8-s-PDLA-15K-5% prepared by different foaming pressures: (**a**) 16 MPa, (**b**) 20MPa and (**c**) 22 MPa at fixed temperature of 120 °C for 2 h.

**Figure 11 polymers-15-01984-f011:**
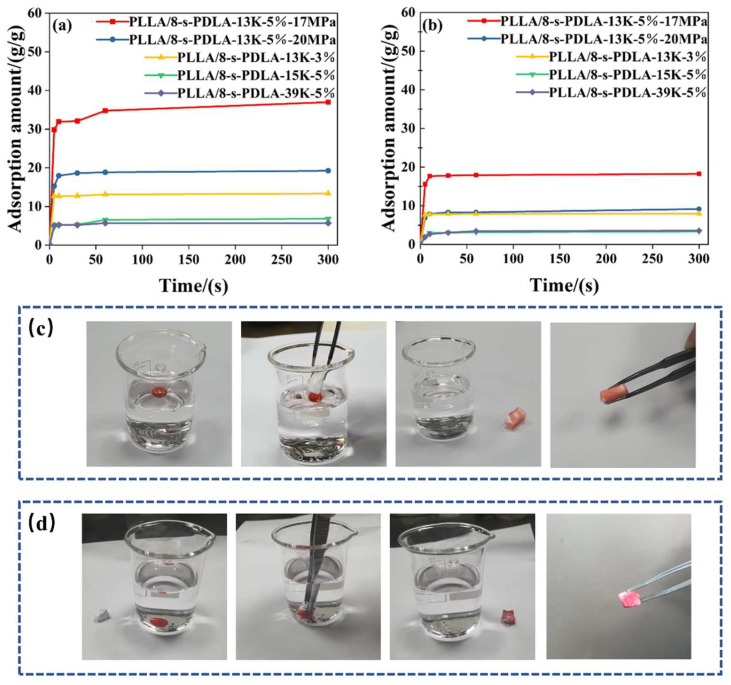
Adsorption capacity of the PLLA/8-s-PDLA foams for (**a**) CCl_4_ and (**b**) silicone oil; related adsorption process for (**c**) CCl_4_ and (**d**) silicone oil.

**Figure 12 polymers-15-01984-f012:**
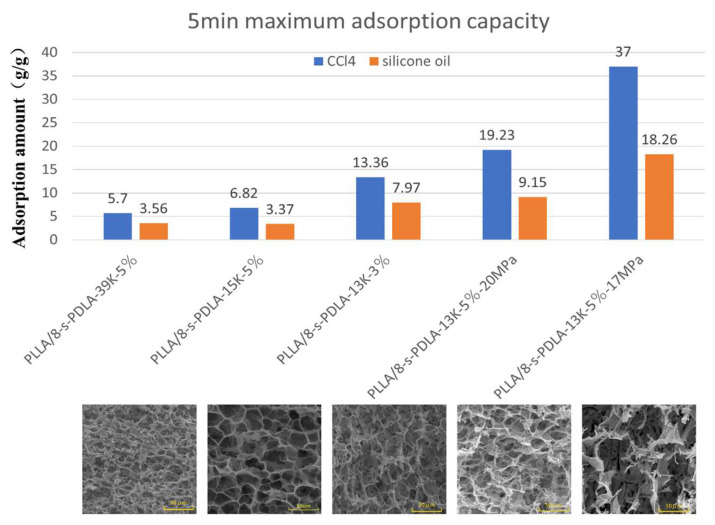
Maximum adsorption capacity of series of PLLA/8-s-PDLA blends for CCl4 in blue column and silicone oil in orange column within 5 min with the corresponding SEM images of foam morphology.

**Table 1 polymers-15-01984-t001:** Characteristic parameters of PLLA and PLLA-s-PDLAs foams.

Specimens	Cell Size(μm)	Density (g/cm^3^)	Expansion Ratio	Cell Density
PLLA	28.26	0.42	3.03	1.93 × 10^8^
PLLA/8-s-PDLA-39K-5%	14.60	0.17	7.59	3.50 × 10^9^
PLLA/8-s-PDLA-15K-5%	27.95	0.15	8.63	5.68 × 10^8^
PLLA/8-s-PDLA-13K-5%	20.52	0.09	14.45	2.40 × 10^9^
PLLA/8-s-PDLA-39K-3%	23.55	0.10	13.50	1.49 × 10^9^
PLLA/8-s-PDLA-13K-5% 17MPa	61.74	0.0532	24.06	1.47 × 10^8^
PLLA/8-s-PDLA-13K-1%+39K-2%	12.43	0.18	7.03	5.26 × 10^9^

**Table 2 polymers-15-01984-t002:** Comparison of the maximum adsorption capacity of the PLLA/8-s-PDLA foams prepared in this work with that of the reported oil-absorption materials in the literature.

Authors	Materials	Expansion Multiplier	Maximum Adsorption Capacity(g/g)	Required Adsorption Time (min)	Paper
Those of this paper	PLLA/s-PDLA	24.1	37.0	<1	
Li et al.	PLA/PBS	43.6	21.9	>15	[19]
Li et al.	PLA	-	4.1	-	[34]
Wang et al.	PLA	59.7	15.0	-	[35]
Wei et al.	PLLA/PTFE	10.2	6.1	>20	[36]
Hua et al.	PLLA/m-LA	40.17	12.4	>5	[37]

## Data Availability

The data are available from the corresponding author upon reasonable request.

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
