# Peer review of "High-Expansion Open-Cell Polylactide Foams Prepared by Microcellular Foaming Based on Stereocomplexation Mechanism with Outstanding Oil–Water Separation"

_polymers, 2023, doi:10.3390/polym15091984_

Round 1

Reviewer 1 Report

Manuscript Title: “High-Expansion Open-Cell Polylactide Foams Prepared by Microcellular Foaming Based on Stereocomplexation Mechanism with Outstanding Oil-Water Separation

In this manuscript, Authors report that Biodegradable polactide (PLA) foams with open-cell structure are good candidate for the oil-water separation and claimed that they used a green facile strategy for the fabrication of open-cell fully biodegradable PLA-based foams is proposed by introducing the unique stereocomplexation mechanism between PLLA and synthesized a series of star-shaped PDLA for the first time. Though the study is interesting, I, therefore, recommend a major revision of this work before further consideration. The major comments are

1.       Why are PLA pellets dried for 12 hours at 60 °C? Vacuum-assisted drying at 60 °C is needed to remove any trapped moisture?

2.       What method was used to recover the catalyst (Sn(Oct)2) after the polymerization process? It’s not clear from the process mentioned.

3.       How the eight arms of the PDLA's 8-star form have been confirmed? How many possible arms a polymer could have can be confirmed by NMR spectroscopy. But authors need to elaborate more about the integration side chain and backbone.

4.       The NMR peaks 3 and 4 could only come from one arm. In NMR spectra, where is the peak of -OH from arm X?

5.       How is the integration of proton from the -CH3 group 6.1 (6) in the arm if the integration of H (one H) is 1? 

6.       Authors should find the relevant literature to confirm the peaks in NMR spectra and site them at the appropriate positions.

7.       Appropriate citations, specifically in the results and discussion section, citations are missing. Writers are encouraged to make sure they have the right citations for any claims they make in the result and discussion sections. In the entire result and discussion section, not a single citation is available. This gives credence to your discussion of findings and make comparison between findings possible.

8.       Were the molecular weights determined by NMR and GPC similar? and what was the significance of checking molecular weight from both techniques?

9.       What are the POM photographs? It should be described in the relevant section and measurements should be described in experiment section.

10.   It is advised to use of the same magnification SEM images. Such as, Figure 6 shows the sample a (1000 X) image at 20 u, whereas samples (b-d) were SEM images of 50 u. Similarly, in Figure 8., 50 u pictures are used rather than 20 u.

11.   Any acronyms (such as PBAT, PLLA, PBS, and so on) throughout the entire manuscript, should be expanded in the first place.

12.   Supporting information is missing.

13.   The whole manuscript should be thoroughly proof-read in order to improve the English language. 

Reviewer 2 Report

The paper reports research results on an interesting route to produce open cell foams from PLA. The exploitation of PLA L and D stereocomplexes is a good way to promote physical crosslinking and compensate for the low melt strength of the polymer. Furthermore, it is interesting the absorption performance of the produced open cell foams.

Some issues arise from the reading of the manuscript.

The first issue is the language. A careful proofreading of the text must be performed. Several sentences present errors and are not clear, just like some words have been cut and pasted.

How do the authors explain the crystallization temperature of ...PDLA-15K-5% sample and ...PDLA-13K-5% samplein fig.3? the intermediate length and content, respectively, should give and intermediate Tc. Instead, it is very different from that of the extremes.

The authors measured the rheological properties of polymers by means of a rotational rheometer. The melt strength cannot be evaluated by this method. Extensional or, at least, capillary rheometry are more appropriate. The authors should avoid to relate the "improved melt strength" to the complex viscosity results they got. 

Furthermore, the improved capability of the blends to withstand the expansion process is related to the development of physical crosslinking points (due to the stereocomplexation of PLLA and PDLA stars).

No data are reported on the CO2 absorption of the developed blends. It is not clear why the presence of SC-PLA should allow an increase in the CO2 solubility. Nor it is demonstrated that SC-PLA is a gas-reservoir. Please add data or support these statements with references.

In figure 9 the cellular morphology is not open, at least not completely open. Some cells show interconnections, but the cellular morphology is very different with respect to the foams in figures 6b,d and figure 8a-d. The same comment applies to figure 10, where the cellular morphology clearly shows cell walls, with few (large) interconnections between cells. I would clarify in the manuscript this.

Round 2

Reviewer 1 Report

Authors have complied all the comments and revised the manuscript accordingly.

Supporting informations is also attached

I recommend accepting the updated manuscript.

Sincere regards

Reviewer 2 Report

Albeit I still have some concerns about the rheology and the cellular morphology, I think that the authors clarified most of my concerns.

The authors stated "In the revised manuscript, we omitted the confused explanation that SC-PLA is a gas-reservoir." but I still see the comment in the conclusions. Since no explanation has been given in the discussion section, I believe that the comment should be removed from the Conclusions paragraph.

The paper can be accepted for the publication after minor revisions.